# Intravenous Polyethylene Glycol Alleviates Intestinal Ischemia-Reperfusion Injury in a Rodent Model

**DOI:** 10.3390/ijms241310775

**Published:** 2023-06-28

**Authors:** Mathias Clarysse, Alison Accarie, Arnau Panisello-Roselló, Ricard Farré, Emilio Canovai, Diethard Monbaliu, Gert De Hertogh, Tim Vanuytsel, Jacques Pirenne, Laurens J. Ceulemans

**Affiliations:** 1Department of Abdominal Transplant Surgery & Transplant Coordination, University Hospitals Leuven, 3000 Leuven, Belgium; mathias.clarysse@kuleuven.be (M.C.); emilio.canovai@ouh.nhs.uk (E.C.); diethard.monbaliu@uzleuven.be (D.M.); jacques.pirenne@uzleuven.be (J.P.); 2Abdominal Transplantation Laboratory, Department of Microbiology, Immunology and Transplantation, KU Leuven, 3000 Leuven, Belgium; 3Leuven Intestinal Failure and Transplantation Center (LIFT), University Hospitals Leuven, 3000 Leuven, Belgium; tim.vanuytsel@kuleuven.be; 4Translational Research Center for Gastrointestinal Disorders (TARGID), Department of Chronic Diseases and Metabolism (CHROMETA), KU Leuven, 3000 Leuven, Belgium; alison.accarie@gmail.com (A.A.); ricard.farre@kuleuven.be (R.F.); 5Institut d’Investigacions Biomèdiques de Barcelona (IIBB), Consejo Superior de Investigaciones Cientificas (CSIC)—Institut D’Investigacions Biomèdique August Pi I Sunyer (IDIBAPS), 08036 Barcelona, Spain; arnau.panisello@iibb.csic.es; 6Department of Pathology, University Hospitals Leuven, 3000 Leuven, Belgium; gert.dehertogh@kuleuven.be; 7Laboratory of Translational Cell & Tissue Research, KU Leuven, 3000 Leuven, Belgium; 8Department of Gastroenterology and Hepatology, University Hospitals Leuven, 3000 Leuven, Belgium; 9Department of Thoracic Surgery, University Hospitals Leuven, 3000 Leuven, Belgium; 10Laboratory of Respiratory Diseases and Thoracic Surgery (BREATHE), Department of Chronic Diseases and Metabolism (CHROMETA), KU Leuven, 3000 Leuven, Belgium

**Keywords:** intestinal ischemia, intestinal ischemia-reperfusion injury, polyethylene glycol

## Abstract

Intestinal ischemia-reperfusion injury (IRI) is a common clinical entity, and its outcome is unpredictable due to the triad of inflammation, increased permeability and bacterial translocation. Polyethylene glycol (PEG) is a polyether compound that is extensively used in pharmacology as an excipient in various products. More recently, this class of products have shown to have potent anti-inflammatory, anti-apoptotic, immunosuppressive and cell-membrane-stabilizing properties. However, its effects on the outcome after intestinal IRI have not yet been investigated. We hypothesized that PEG administration would reduce the effects of intestinal IRI in rodents. In a previously described rat model of severe IRI (45 min of ischemia followed by 60 min of reperfusion), we evaluated the effect of IV PEG administration at different doses (50 and 100 mg/kg) before and after the onset of ischemia. In comparison to control animals, PEG administration stabilized the endothelial glycocalyx, leading to reduced reperfusion edema, bacterial translocation and inflammatory reaction as well as improved 7-day survival. These effects were seen both in a pretreatment and in a treatment setting. The fact that this product is readily available and safe should encourage further clinical investigations in settings of intestinal IRI, organ preservation and transplantation.

## 1. Introduction

Intestinal ischemia is a common and life-threatening condition that often has an insidious initial presentation and a high mortality rate [1,2]. Even after reperfusion, damage continues through an inflammatory process known as ischemia-reperfusion injury (IRI) [3]. This inflammatory process is caused by a complex array of detrimental components such as Adenosine Triphosphate (ATP) depletion, mitochondrial calcium overload and Reactive Oxygen Species (ROS) formation [4]. This detrimental process activates the innate immune system, provoking local and systemic inflammation which leads to cell death [5,6,7]. In the intestine, a single layer of tightly linked epithelial cells forms a barrier between the intestinal lumen and underlying innate immune cells [8]. In this context, IRI quickly leads to the disruption of this structure, resulting in the translocation of endotoxins and bacterial products.

Polyethylene glycols (PEGs) are polyether compounds that are water-soluble polymers with variable molecular weight. They are used extensively in many applications, most frequently being used as excipients for many different pharmaceutical products, laxatives, creams and lubricants [9,10]. More recently, PEG has been shown to have anti-inflammatory, anti-apoptotic, immunosuppressive and endothelia-cell-membrane-stabilizing properties [11,12,13,14,15,16,17]. For example, intravenous administration (IV) of high-molecular-weight PEG (35 kDa) reduced the IRI of hearts and livers and improved the preservation of steatotic livers in preclinical studies [17,18,19,20]. These beneficial effects were partly due to the activation of pro-survival pathways (protein kinase B (Akt), adenosine monophosphate protein kinase (AMPK) and endothelial nitric oxide synthase (eNOS)) [10]. PEG also stabilizes the actin cytoskeleton in endothelial cells through sarcolemmal lipid-raft architecture preservation [17,21]. However, there are, to our knowledge, no studies on the effect of PEG on intestinal IRI. The aim of this study was therefore to investigate the protective role of PEG in a rodent model of warm intestinal IRI.

## 2. Results

### 2.1. Intravenous PEG Pretreatment Did Not Prevent Epithelial Intestinal Damage but Reduced Paracellular Permeability

In our rodent model of 45 min of atraumatic clamping of the superior mesenteric artery, IRI had a clear impact on the epithelial injury of the distal ileum. Epithelial damage, measured in histological samples by Park–Chiu score after IRI, was 5.0 ± 1.1, in contrast to 0 in the sham condition (*p* = 0.009). However, pretreatment with increasing dosages of PEG did not alter this outcome: 4.7 ± 1.8 for PEG50 + IRI (50 mg/kg PEG) (*p* = 1 versus saline + IRI) and 3.0 ± 2.0 for PEG100 + IRI (100 mg/kg PEG) (*p* = 1 versus saline + IRI) (Figure 1A). The observation was confirmed for the villus length, measuring 295.2 ± 20.9 µm for sham, 113.2 ± 31.4 µm for saline + IRI (*p* < 0.001 versus sham), 101.5 ± 9.9 µm for PEG50 + IRI (*p* = 0.97 versus saline + IRI) and 136.0 ± 49.5 µm for PEG100 + IRI (*p* = 0.76 versus saline + IRI) (Figure 1B). These histopathologic alterations were reflected by an increased epithelial permeability, as studied by measuring the transepithelial electrical resistance (TEER) in an Ussing chamber setup: 32.1 ± 4.6 Ω × m^2^ in sham versus 7.0 ± 1.9 Ω × cm^2^ in the saline + IRI group (*p* = 0.003), as well as 7.1 ± 1.5 Ω × cm^2^ in PEG50 + IRI (*p* = 1), and 11.1 ± 5.5 Ω ×cm^2^ in PEG100 + IRI (*p* = 1) (Figure 1C). Increased systemic LPS levels (endotoxin) confirmed this increased permeability with 0.009 ± 0.002 U/mL plasma endotoxin in sham versus 0.116 ± 0.118 U/mL in the saline + IRI group (*p* = 0.004), 0.065 ± 0.030 U/mL in PEG50 + IRI (*p* = 1) and 0.027 ± 0.009 U/mL in PEG100 + IRI (*p* = 0.97) (Figure 1D).

### 2.2. High-Dose PEG Pretreatment Reduced the Inflammatory Response

Following high-dose PEG pretreatment (100 mg/kg), plasma levels of interleukin IL-6 significantly decreased in contrast to control intestinal IRI (saline + IRI): 233.8 ± 81.8 pg/mL vs. 685.7 ± 366.9 pg/mL (*p* = 0.03) (Figure 2A). The protective effect of PEG pretreatment was also measurable at the gene transcription level in tissues. Tissue IL-1β levels were significantly altered after IRI: 1.0 ± 0.3 fold change (sham) vs. 10.2 ± 3.1 fold change (saline + IRI) (*p* < 0.001). This inflammatory reaction was significantly reduced after PEG pretreatment (PEG100 + IRI): 5.4 ± 2.4 fold change (*p* = 0.004) (Figure 2B). When looking at tumor necrosis factor—alfa (TNF-α), PEG pretreatment moderately, however not significantly, toned down the inflammatory response: 6.8 ± 2.3 fold change (saline + IRI) vs. 5.1 ± 1.4 fold change (PEG100 + IRI) (*p* = 0.32) (Figure 2C). Significantly lower levels of the anti-inflammatory cytokine IL-10 were observed: 24.1 ± 17.1 fold change (saline + IRI) vs. 8.0 ± 4.0 fold change (PEG100 + IRI) (*p* = 0.02) (Figure 2D). Interferon—gamma (IFN-γ) levels were increased following intestinal IRI: 1.0 ± 0.4 fold change (sham) vs. 5.7 ± 2.6 fold change (saline + IRI) (*p* = 0.02), but not significantly reduced following PEG pretreatment: 2.4 ± 1.7 fold change (PEG100 + IRI) (*p* = 0.17) (Figure 2E).

### 2.3. High-Dose PEG Pretreatment Preserved Vascular Permeability by Protecting the Endothelial Glycocalyx

At the moment of sacrifice, there was less hemoconcentration seen after PEG pretreatment in comparison to the control intestinal IRI: 17.2 ± 1.8 g/dL (saline + IRI) vs. 13.2 ± 1.0 g/dL (PEG100 + IRI) (*p* = 0.001) (Figure 3A, hemoglobin). This finding was confirmed by the reduced reperfusion edema (wet/dry ratio) of the intestinal tissue: 6.5 ± 0.9 (saline + IRI) vs. 4.2 ± 0.6 (PEG100 + IRI) (*p* < 0.001) (Figure 3B). This effect could not be explained by a difference in the osmolality of the infused solutions: 319 mOsm/kg (saline + IRI) vs. 316 mOsm/kg (PEG100 + IRI). By analyzing the endothelial glycocalyx, the plasma levels revealed a reduced shedding of its main components syndecan-1—126.0 ± 28.0 ng/mL (saline + IRI) vs. 92.3 ± 14.4 ng/mL (PEG100 + IRI) (*p* = 0.11)—and heparan sulfate—190.1 ± 82.6 ng/mL (saline + IRI) vs. 25.6 ± 34.1 ng/mL (PEG100 + IRI) (*p* = 0.02) (Figure 3C,D). This protective effect on the endothelial glycocalyx of the intestine was confirmed by transmission electron microscopy (TEM) images showing an intact endothelial glycocalyx and underlying structures after PEG pretreatment (Figure 3E) vs. the destruction of both in the case of intestinal IRI without PEG pretreatment (Figure 3F).

### 2.4. Treatment with High-Dose PEG Improved Outcome after Intestinal IRI

When 100 mg/kg of PEG was administered 15 min after the onset of ischemia as a treatment option (IRI + PEG100(T)), no significant difference was seen in the epithelial histopathologic injury in comparison with controls (saline + IRI) or high-dose pretreatment (PEG100 + IRI): 5.0 ± 0.0 (IRI + PEG100(T)) vs. 5.0 ± 1.1 (saline + IRI) (*p* = 1), or vs. 3.0 ± 2.0 (*p* = 1), respectively (Figure 4A). Nor was this the case for bacterial translocation (Figure 4B). Regarding inflammation, tissue IL-1β was decreased in the PEG-treated animals: 10.2 ± 3.1 fold change (saline + IRI) vs. 4.7 ± 1.5 fold change (IRI + PEG100(T)) (*p* = 0.001) (Figure 4C). It was also decreased in tissue: IL-10: 24.1 ± 17.1 fold change (saline + IRI) vs. 7.7 ± 4.0 fold change (IRI + PEG100(T)) (*p* = 0.02) (Figure 4D). Reduced reperfusion edema was also seen with PEG: 6.5 ± 0.9 (saline + IRI) vs. 4.9 ± 0.5 (IRI + PEG100(T)) (wet/dry ratio) (*p* < 0.001) (Figure 4E). However, no significant difference was seen on plasma syndecan-1 levels: 125.9 ± 28.0 ng/mL (saline + IRI) vs. 108.9 ± 23.5 ng/mL (IRI + PEG100(T)) (*p* = 0.69) (Figure 4F). Complete graphs of all analyses performed on all investigated groups can be found in the Appendix A (Appendix A).

### 2.5. IV PEG Administration Improved Survival after Intestinal IRI

A seven-day survival of 40% was observed in our rodent model of 45 min of warm intestinal ischemia. A dose effect was seen in the pretreatment setting, with 70% survival for 50 mg/kg PEG and 80% for 100 mg/kg PEG. Administering PEG as a treatment increased the 7-day survival up to 70% (*p* = 0.0117) (Figure 5).

## 3. Discussion

In this study on rodent intestinal IRI, we investigated the effect of IV PEG administration, in both the pretreatment and treatment settings. In both cases, IV 35 kDa PEG improved the outcome after intestinal IRI by reducing the vascular permeability, as observed by a decreased shedding of the main components of the endothelial glycocalyx (syndecan-1 and heparan sulfate) and superior structural preservation, as shown by TEM, despite the ongoing damage to the intestinal epithelium. Secondary effects of reduced vascular permeability were demonstrated by a decreased hemoglobin concentration and a reduction in reperfusion edema, inflammatory reaction as well as bacterial translocation.

The endothelial glycocalyx is the luminal lining of vascular endothelial cells with membrane-bound proteoglycans, glycosaminoglycans and sialic-acid-containing glycoproteins [22]. Destruction of the endothelial glycocalyx has recently been proposed as the earliest form of structural damage and a main pathophysiologic mechanism of the detrimental effects of IRI [22,23]. While previously demonstrated in other organs, the present study is the first, to our knowledge, to analyze this mechanism in the intestine. Previous studies, looking at both static and dynamic liver preservation, demonstrated that PEG-containing preservation solutions helped to stabilize the endothelial glycocalyx by PEGylation, which prevented breakdown and led to reduced preservation damage whilst improving nitrous oxide production [10,20,24,25]. The close proximity of PEG to the endothelial glycocalyx, and also the protective effect on the underlying endothelial cells, was shown by Chiang et al. in immunofluorescent studies [21]. The administration of PEG protected the human lung endothelium by inducing significant cytoskeletal rearrangement through the formation of well-defined cortical actin rings and lamellipodia. These lamellipodia contain actin-binding proteins which are important for cell–matrix and cell–cell junctional adhesion [21]. In other words, PEGylation of protein kinases and the endothelial glycocalyx stabilized the endothelial glycocalyx and reduced IRI-induced damage. PEG’s ability to maintain the endothelial cytoskeleton and its effect on the associated tight junctions and paracellular permeability has been proven previously in liver IRI and preservation studies [17,21,25]. Furthermore, PEG has previously been shown to enable mitochondrial protection in the face of IRI by activating pro-survival kinase Akt and cytoprotective factor AMPK, as well as via apoptosis inhibition [13,17,19,25]. As such, the 7-day survival benefit with PEG administration could be explained by this endothelial glycocalyx modification. Mortality after intestinal IRI is mainly due to the increased intestinal epithelial permeability, secondary bacterial translocation, systemic inflammatory response syndrome and, hence, multi-organ failure [26]. According to our findings, PEG administration significantly improved survival by protecting against bacterial translocation and systemic inflammatory response syndrome. We believe that PEG allowed the animals to survive long enough for the small bowel to regenerate. This was supported by our finding that at the 7-day sacrifice interval, the bowel looked completely macro- and microscopically normal.

Some of the protective effects described above (hemoglobin concentration and reperfusion edema) could also be attributed solely to the oncotic effect of the PEG solution, which is independent of any particular PEG-specific mechanism. This is one of the reasons for its current use in several organ preservation solutions [27,28,29]. This potentially suppressed endothelial cell swelling and other changes in vascular cells were already attributed to the colloid in these preservation solutions by Belzer and Southard 30 years ago [30,31]. This potential oncotic effect could have been tested by the addition of an extra test group with oncotic active agents, such as albumin for example. This might be of interest in further studies. However, osmotic changes induced by the differences in preservation solutions could be mainly attributed to other components (raffinose and lactobionic acid) than the colloid itself [31,32]. However, despite the presence of the PEG in the infused solution in this study, the osmolality was similar to the control saline solution. Therefore, the protective effect could not simply be explained by this mechanism only and might be explained by the more specific effects on the endothelial glycocalyx, as described above.

IV administration of PEG could not protect the intestine against injury to the epithelial lining, as shown by an increased Park–Chiu score and lowered villus length. This is in contrast with studies investigating the luminal administration of PEG. Both in experimental and clinical settings, it has been shown that luminal PEG enhances intestinal preservation injury by protecting against mucosal damage and redistributing tight-junction proteins [14,33,34,35]. As such, future experiments on both IV and intraluminal PEG administration should be performed to study a potential synergistic effect that could further improve the outcome after intestinal IRI, organ preservation and intestinal transplantation.

PEG exists in different molecular sizes. However, in this study, a high-molecular-weight PEG of 35 kDa was chosen as it is known to be a safe component of the preservation solution IGL-1 [29]. Secondly, PEG molecules of 20–40 kDa are slowly eliminated by the kidneys or in the feces and, hence, pharmacokinetic studies have shown adequate plasma levels in the plasma of rodents up to 12 weeks after single IV administration [10,36]. As such, PEG can exert its protective effects not only at the immediate onset of intestinal IRI, but also on secondary remote organ injury provoked by IRI [37].

This experimental study has its limitations. First, structural changes were shown in the endothelial glycocalyx, in combination with secondary, indirect effects. However, whether these structural changes have direct mechanistic or pathophysiologic advantages was not studied here. Secondly, an additional test group with oncotic active agents, for example albumin, could have tested the potential effect of the oncotic pressure difference between the infused solutions. Thirdly, the epithelial glycocalyx, mainly consisting of several mucin proteins, was not studied here, as it was not the focus of the study. PEG was only administered intravenously and no differences were measured on histopathologic examination or epithelial permeability. Although we did not study it specifically, we do not suspect any differences at the epithelial glycocalyx level. A completely destroyed intestinal epithelium, as shown with the histopathologic scoring by Park–Chiu, is unlikely to leave much epithelial glycocalyx. On the other hand, studies by Oltean et al. showed that enteral administration of PEG had a protective effect on the preservation of the intestinal epithelium [33,35]. In this setting, it might be interesting to have a look into the epithelial glycocalyx of the small bowel as well.

The direct availability of PEG and its current role in clinical practice pave the way for further clinical investigation, for example by comparing intraluminal and vascular PEG administration in the setting of intestinal IRI. In the case of a patient presenting with acute intestinal ischemia, multimodal management strategies, such as those proposed by Corcos et al. and those consisting of oral and systemic antibiotics, proton-pump inhibitors, resuscitation, etc., could be started [38]. IV PEG could be added to this multimodal management strategy and alter the outcome in these patients by preventing secondary bacterial translocation and systemic inflammatory response syndrome. On the other hand, in the setting of intestinal transplantation with expected IRI, IV PEG could be administered during donor management preprocurement and at the recipient pretransplant stage as a pretreatment modulation. Furthermore, PEG could also be added in the preservation solutions, as next to its oncotic effect it might have additional beneficial effects by protecting the vascular endothelium.

## 4. Materials and Methods

### 4.1. Animal Model

Male Sprague Dawley rats (Janvier Labs, Saint Berthevin Cedex, Mayenne, France), weighing 275–350 g, 6 weeks of age, were housed in the KU Leuven animal facility under specific pathogen-free conditions, with 2–3 per cage. The rats were acclimatized for 5–7 days before any intervention. The animals were kept in 14/10 h light/dark cycles, controlled temperature, and received rat chow and water ad libitum. The animals were not fasted before surgery. Institutional animal research ethical committee approval—following the EU directive for animal experiments—was granted under the number P120/2016. The reported animal study complies with the ARRIVE guidelines 2.0 [39].

### 4.2. Anesthesia

Animals were anesthetized by an intraperitoneal (I.P.) injection mix of 57.14 mg/kg body weight (BW) of ketamine (100 mg/mL, Nimatek, Eurovet, Bladel, The Netherlands) and 5.71 mg/kg BW of xylazine (Xyl-M 2%, Inovet, Arendonk, Belgium). Following animal welfare standards, rats were monitored at least 3 times daily and buprenorphine subcutaneous (0.016 mg/kg BW, 0.3 mg/mL, Vetergesic, Ceva, Brussel, Belgium) was used for analgesia, twice daily, during the first 3 days following the surgery [26,40].

### 4.3. Surgery

IRI was performed with 45 min of ischemia and 60 min of reperfusion. Intestinal IRI was induced, after median laparotomy of 4 cm on the linea alba, by isolated atraumatic clamping of the superior mesenteric artery. Ischemia was checked by pulselessness in the mesentery and discoloration/dysmotility of the bowel. The laparotomy wound was temporarily closed during the experiment. At reperfusion, 1 mL of warmed saline (37 °C) was administered intraperitoneally to compensate for fluid loss by evaporation. Reperfusion was checked by recovery of arterial pulsations in the mesentery, recoloration and regain of motility. All experiments were performed by the same researcher (M.C.). The laparotomy wound was closed subcutaneously in 2 layers with Prolène 4-0 (Ethicon, Machelen, Belgium), and 0.5 mL of ropivacaine (3.16 mg/kg BW, 2 mg/mL, Naropin, Aspen, Ireland) was administered in the wound edges for local analgesia.

An equal volume of vehicle (NaCl 0.9%) or PEG (35 kDa, dissolved in NaCl 0.9%, provided by the Institute of Biomedical Research, Barcelona (IIBB-CSIC, Catalonia, Spain)) was administered intravenously 10 min before (pretreatment) or 15 min after (treatment) ischemia onset, by injection in the penile vein. PEG was administered in increasing dosages (50 and 100 mg/kg BW). These dosages were based upon preliminary dose-dependent survival and toxicity studies, of which the results can be seen in Appendix A and Appendix A.

For the 60 min reperfusion experiments, rats were randomly divided into five groups (*n* = 6/group) (Table 1, Figure 6):

Survival analysis was performed with ten animals per group and observed on a 3-times-daily basis for 7 days.

At the end of the experiment, all animals were sacrificed by exsanguination, followed by blood and intestinal tissue collection. In the 60 min reperfusion group, the animals were under anesthesia until exsanguination. In the survival group, rats were anesthetized with pentobarbital before sacrifice (65 mg/kg BW, 200 mg/mL, Dolethal, Vetoquinol, Niel, Belgium).

### 4.4. Blood and Tissue Sampling

Heparinized blood samples were collected after puncture of the aorta for blood gas analysis (0.4 mL) into 2 EDTA tubes. The tubes were spun at 3500 rpm for 10 min at 4 °C. Plasma was snap-frozen in liquid nitrogen and stored at −80 °C. One ileal tissue sample of 5 cm was taken just proximally of the ileocecal valve, kept in glucose buffer on ice and mounted in the Ussing chambers. Ileal samples were collected proximally to the previous and preserved in 4% buffered formalin for histological evaluation and snap-frozen, after feces removal, for molecular analysis.

### 4.5. Arterial Blood Gas Analysis

Hemoglobin levels (g/dL) and hematocrit (%) were analyzed by a blood gas analyzer (ABL-815, Radiometer, Brønshøj, Denmark).

### 4.6. Histological Evaluation

Full-thickness samples were formalin-fixed, paraffin-embedded, cut into 5 µm thick sections and stained with hematoxylin-eosin. The ischemic injury was scored in a blinded fashion by an experienced pathologist (G.D.H.) on four fields per section by the usage of the Park–Chiu score [41,42].

Villus length—defined as the distance between the mouth of the crypts and the tip of the villi—was measured in 4 different fields per tissue section, and the average was calculated to avoid the potential impact of patchy necrosis.

### 4.7. Ussing Chamber Experiments

#### Electrophysiological Parameters

Full-thickness ileal tissue (mucosa, submucosa, muscular layer and serosa) was mounted, in triplicate, in a standard vertical Ussing chamber (Mussler Scientific Instruments, Aachen, Germany) with an opening of 9.60 mm^2^ by a blinded, experienced researcher (A.A.). Each half chamber was filled with 3 mL Krebs solution with 10 mM mannitol at the mucosal side, and 10 mM glucose at the serosal side. Both buffers were maintained at 37 °C and continuously oxygenated with 95%/5% O_2_/CO_2_ and stirred by gas flow in the chambers. In this setup, data sampling and pulse inductions were computer-controlled using Clamp software (Version 9.00, Mussler Scientific Instruments, Aachen, Germany). Transepithelial electrical resistance (TEER) was measured by averaging 90 min of measurement, after initial 30 min stabilization. All tissues were mounted within 10 min after the exsanguination of the animal.

In our particular setting of intestinal IRI leading to diminished/denudated mucosal surface area (as shown by Grootjans et al. [43]), TEER was corrected by multiplying TEER with its corresponding villus length divided by the average villus length of the sham group [26].

### 4.8. Bacterial Translocation

Quantification of plasma endotoxin levels was performed by the colorimetric Limulus amebocyte lysate test (LAL QCL-1000^TM^, Lonza, Bornem, Belgium), according to the manufacturer’s instructions. Absorbance was measured spectrophotometrically by FLUOstar Omega (BMG Labtech, Ortenberg, Germany) at 410 nm. Corrections were made by the subtraction of the absorbance of the sample without the addition of LAL.

### 4.9. Edema

Tissue water content (edema) was assessed by the ratio between the weight before and after drying. Snap-frozen, whole-thickness ileal tissue samples were weighed just before and immediately after drying them for 3 h at 80 °C in a drying oven with forced convection (VENTI-Line VL 115, VWR, Leuven, Belgium). The results are expressed as the wet/dry ratio.

### 4.10. Osmolality

The osmolality of each injected solution was determined by using a large-volume osmometer that uses the freezing point depression technique and requires a 500 μL sample (OsmoStation, Menarini Diagnostics, Diegem, Belgium). Each solution’s osmolality was measured twice on the large-volume osmometer.

### 4.11. ELISA

Plasma IL-6 concentration was measured by enzyme-linked immunosorbent assay (ELISA), according to the manufacturer’s instructions (R6000B, Bio-Techne Ltd., Abingdon, UK).

Plasma Syndecan-1 (E-EL-R0996, Elabscience, Houston, TX, USA) concentration was measured by ELISA, according to the manufacturer’s instructions. 

Plasma heparan sulfate (OKEH02552, Aviva Systems Biology, San Diego, CA, USA) concentration was measured by ELISA, according to the manufacturer’s instructions.

### 4.12. Quantitative Reverse-Transcription Polymerase Chain Reaction (qRT-PCR)

The relative expression of pro-inflammatory cytokines (interleukin (IL)-1β, tumor necrosis factor (TNF)-α, interferon (IFN)-γ) and anti-inflammatory cytokine (IL-10)) were determined by qRT-PCR. Tissue was homogenized in TRIzol reagent (Life Technologies, Carlsbad, CA, USA) and total ribonucleic acid (total RNA) was extracted using the RNeasy isolation kit (Qiagen, Maryland, MD, USA) according to the manufacturer’s instructions. cDNA was synthesized from 200 ng total RNA using M-MLV transcriptase (Life Technologies, Carlsbad, CA, USA). Next, a real-time PCR reaction was performed on a LightCycler 96 W (Roche, Vilvoorde, Belgium) with Taqman Fast Universal PCR Master Mix and Taqman gene expression assays (Applied Biosystems, Life Technologies, Carlsbad, CA, USA) (IL-1β (Rn00580432_m1), TNF-α (Rn00562055), IFN-γ (Rn00594078) and IL-10 (Rn00563409)). A three-step amplification program was used: 95 °C for 10 min, followed by 45 cycles of amplification (95 °C for 10 s, 60 °C for 15 s, 72 °C for 10 s). Target messenger RNA (mRNA) expression for cytokines was quantified relative to the housekeeping gene GAPDH (Life Technologies, Carlsbad, CA, USA) and to the controls using the “−ΔΔCt method”.

### 4.13. Transmission Electron Microscopy (TEM)

Concerning TEM imaging, additional rats were perfused with a mixture of 4% paraformaldehyde and 0.2% glutaraldehyde in PBS, via infusion in the left cardiac ventricle, accompanied by a small incision in the right atrium to allow fluctuation of the solution. Tissues were cut into small pieces and fixed with 2.5% glutaraldehyde and 2% paraformaldehyde in 0.1 M phosphate buffer. Samples were postfixated with osmium tetroxide and dehydrated with acetone, embedded in Spurr resin and sectioned using Leica ultramicrotome UC7 (Leica Microsystems, Machelen, Belgium). Ultrathin sections (50–70 nm) were stained with 2% uranyl acetate for 10 min, a lead-staining solution for 5 min and then analyzed with a transmission electron microscope, JEOL JEM-1010 fitted with a Gatan Orius SC1000 (model 832) digital camera at the TEM-SEM Electron Microscopy Unit, Scientific and Technological Centers of the University of Barcelona (CCiTUB).

### 4.14. Statistical Analysis

All data are expressed as mean ± standard deviation and represented in scattered plots. The line in the middle is plotted at the mean. The whiskers indicate the standard deviation. Data were checked for outliers by the ROUT method with Q = 1% and subjected to (log)normality testing (Shapiro–Wilk test). Comparisons between multiple groups were performed with one-way ANOVA and post hoc Tukey’s test in the case of a normal distribution or the Kruskal–Wallis and post hoc Dunn tests for non-normal distribution. Survival analysis was performed by the Kaplan–Meier test (log-rank test). A *p*-value < 0.05 was considered statistically significant (GraphPad Prism version 9.5.1 for Windows, GraphPad Software, San Diego, CA, USA).

## 5. Conclusions

PEG is a non-toxic, water-soluble polymer, and IV administration of high-molecular-weight PEG (35 kDa), either as pretreatment or treatment in the setting of intestinal IRI, has shown anti-inflammatory, immunosuppressive and cell-membrane-stabilization effects. These effects may be partly related to the stabilization of the endothelial glycocalyx and its secondary effects. The abundant availability of PEG in current clinical practice paves the way for further clinical investigations on intestinal IRI and organ preservation and transplantation.

## Figures and Tables

**Figure 1 ijms-24-10775-f001:**
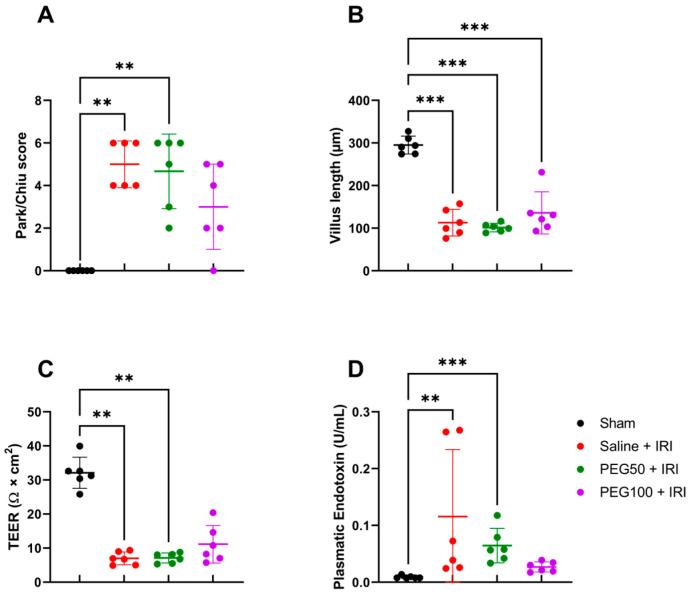
Histopathological effect of PEG pretreatment in intestinal IRI was scored according to the Park–Chiu score (**A**) and villus length (**B**). Intestinal epithelial permeability was measured by TEER in an Ussing chamber, which was corrected for villus length (**C**). Bacterial translocation was evaluated by plasmatic endotoxin levels (**D**). (*n* = 6/group). Statistical analyses were performed by Kruskal–Wallis testing. IRI: ischemia-reperfusion injury; PEG: polyethylene glycol; TEER: transepithelial electrical resistance. ** *p* < 0.01; *** *p* < 0.001.

**Figure 2 ijms-24-10775-f002:**
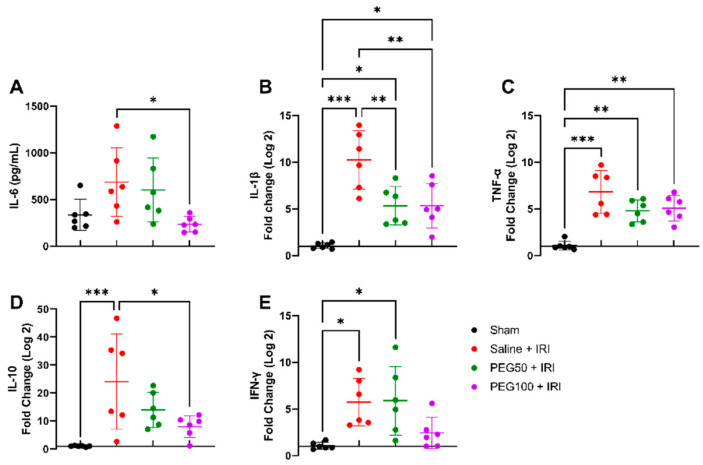
Inflammatory modulation by PEG administration was measured by systemic IL-6 (**A**) and intestinal IL-1β (**B**), TNF-α (**C**), IL-10 (**D**) and IFN-γ (**E**). (*n* = 6/group). Statistical analyses were performed by one-way ANOVA. IFN-γ: interferon-gamma; IL: interleukin; IRI: ischemia-reperfusion injury; PEG: polyethylene glycol; TNF-α: tumor necrosis factor—alfa. * *p* < 0.05; ** *p* < 0.01; *** *p* < 0.001.

**Figure 3 ijms-24-10775-f003:**
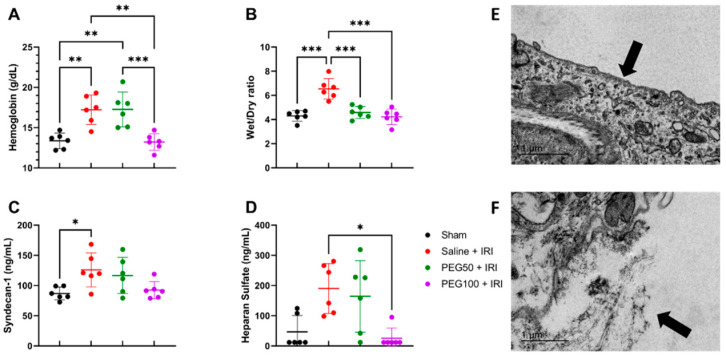
Vascular permeability was altered by IV PEG administration, as shown by hemoglobin levels (**A**), reperfusion edema (wet/dry ratio) (**B**) and plasmatic endothelial glycocalyx components: syndecan-1 (**C**) and heparan sulfate (**D**). (*n* = 6/group). The protective effect was confirmed by TEM images of terminal ileum vasculature of a PEG100 + IRI rat (**E**), compared to a saline + IRI rat (**F**), showing an intact endothelial glycocalyx (arrow **E**) versus a destructed one (arrow **F**). Statistical analyses were performed by one-way ANOVA. IRI: ischemia-reperfusion injury; PEG: polyethylene glycol; TEM: transmission electron microscopy. * *p* < 0.05; ** *p* < 0.01; *** *p* < 0.001. Scale bar = 1 µm.

**Figure 4 ijms-24-10775-f004:**
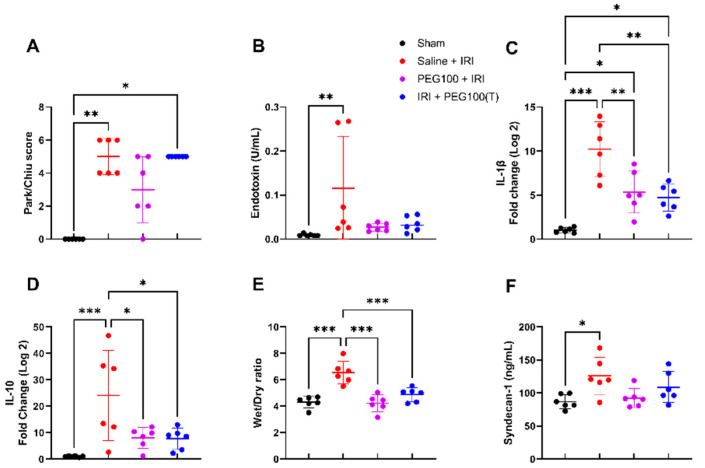
Effect of PEG 100 mg/kg in treatment versus pretreatment and control animals, evaluated on histopathology (**A**), bacterial translocation (**B**), inflammatory reaction (intestinal IL-1β (**C**) and IL-10 (**D**)), reperfusion edema (**E**) and endothelial glycocalyx (**F**). (*n* = 6 rats/group). Statistical analyses were performed by Kruskal–Wallis (**A**,**B**) and one-way ANOVA (**C**–**F**). IRI: ischemia-reperfusion injury; PEG: polyethylene glycol. * *p* < 0.05; ** *p* < 0.01; *** *p* < 0.001.

**Figure 5 ijms-24-10775-f005:**
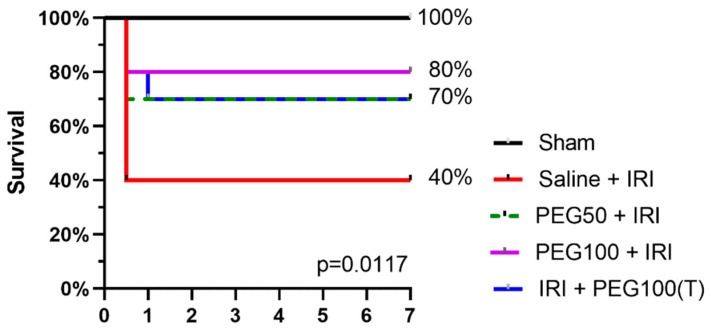
Seven-day survival, assessed by Kaplan–Meier analysis (*n* = 10 rats/group). IRI: ischemia-reperfusion injury; PEG: polyethylene glycol; T: treatment.

**Figure 6 ijms-24-10775-f006:**
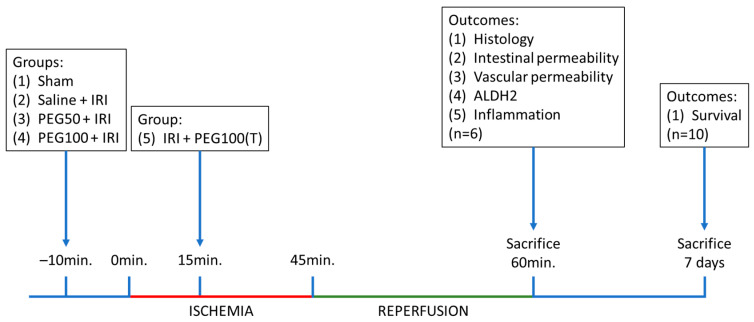
Timeline of the experiments, with sampling time points and explanation of the different groups. IRI: ischemia-reperfusion injury; PEG: polyethylene glycol; 50: 50 mg/kg body weight; 100: 100 mg/kg body weight; 100(T): 100 mg/kg body weight in the treatment setting.

**Table 1 ijms-24-10775-t001:** Experimental groups.

Group	Product	Dosage (mg/kg)	Pretreatment/Treatment
Sham	Saline	0 mg/kg	NA
Saline + IRI	Saline	0 mg/kg	Pretreatment
PEG50 + IRI	PEG	50 mg/kg	Pretreatment
PEG100 + IRI	PEG	100 mg/kg	Pretreatment
IRI + PEG100(T)	PEG	100 mg/kg	Treatment

## Data Availability

All data are available upon request from the corresponding author.

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
