# Peer review of "Intravenous Polyethylene Glycol Alleviates Intestinal Ischemia-Reperfusion Injury in a Rodent Model"

_ijms, 2023, doi:10.3390/ijms241310775_

Round 1

Reviewer 1 Report

I have reviewed the revised manuscript ijms-2467371 entitled “Intravenous polyethylene glycol alleviates intestinal ischemia-reperfusion injury in a rodent model" submitted to the Special Issue of IJMS “Ischemia Reperfusion Injury: A Cell Signaling Crossroads and Therapeutics”, by Dr. Mathias Clarysse et al.

The authors investigated the efficacy of IV administration of PEG in in vivo warm ischemia and reperfusion of rat small intestine. In the treatment group (PEG100+IRI), the 7-day survival rate, tissue qPCR for inflammatory cytokines and chemokines, plasma concentration, edema, vascular dehydration, and leakage of glycocalyx components were significantly reduced, but the tissue damage score did not show significant improvement. These results indicate the importance of the preservation of vascular endothelial glycocalyx for the maintenance of mucosal function, thus, reducing bacterial translocation, edema, and inflammatory response.

The experiments are conducted in a straightforward fashion. The scientific rational and the clinical impact are sound. All in all, the data provided here will be of interest to readers of this journal. The authors need to be addressed some issues.

I have the following concerns.

Major

1)     I understand the intention to present the effective concentration range and administration method of PEG separately, but there is a problem with the way of the data presentation. Figure 4 contains inconclusive data that cannot provide a definite conclusion. The data of PEG50+IRI and IRI+PEG100(T) groups should either be incorporated into other figures or presented as a supplemental figure, considering it as incomplete preliminary study data.

If you want to present Fig.4 with independent significance, additional experiments will be necessary. In other words, it is necessary to examine and present the dose-dependency, toxic and optimal dosage. The impact of differences in administration methods at the optimal dosage should be examined thereafter.

Minor

2)     It is necessary to indicate in the figure legend which statistical analysis method was applied. It would be advisable to use a non-parametric test in case of n=6 each.

3)     This finding indicates the significance of vascular endothelial dysfunction in small intestinal ischemia-reperfusion injury (IRI), and the data on the glycocalyx of the endothelium is highly valuable. It is important that the study focused on the changes in the glycocalyx of the vascular endothelium rather than the small intestinal epithelium. It would be interesting to know the status of the glycocalyx in the small intestinal epithelium. Please discuss about the glycocalyx of endothelial and epithelial cells of small intestine.

4)     In the PEG100+IRI group, tissue qPCR for inflammatory cytokines and chemokines, plasma concentration, edema, vascular dehydration, and leakage of glycocalyx components were significantly reduced, but the tissue damage score did not show significant improvement. Please discuss the reason why survival rates improve despite the lack of improvement in tissue damage. What is the direct cause of death; dysfunction of small intestine or others (shock, acute lung injury and so on)?

Reviewer 2 Report

In their study, Clarysse M et al examined the protective effects of iv PEG against intestinal ischemia/reperfusion injury (IRI) in the anesthetized rat model.

IRI was performed as superior mesenteric artery occlusion, followed by 60 min reperfusion. PEG (50 mg/kg or 100 mg/kg) was infused either before ischemia or 15 min after ischemia. Variables measured were: Transepithelial electrical resistance (TEER), systemic and local inflammation, intestinal permeability, endothelial glycocalyx, intestinal histopathology and transmission electron microscopy, survival.

The results obtained showed protective effects elicited by PEG against inflammation, reperfusion edema, bacterial translocation, and 7-day survival.

The study is interesting and well conducted. The experimental procedures are robust and the conclusions are supported by the results.

I would suggest to improve the discussion about the clinical implications of findings. Which could be the clinical use of PEG? In which clinical condition? 

The quality is good
